# Hyperuricemia during Pregnancy Leads to a Preeclampsia-Like Phenotype in Mice

**DOI:** 10.3390/cells11223703

**Published:** 2022-11-21

**Authors:** Benjamin P. Lüscher, Andreina Schoeberlein, Daniel V. Surbek, Marc U. Baumann

**Affiliations:** 1Department of Obstetrics and Gynecology, Inselspital, Bern University Hospital, 3010 Bern, Switzerland; 2Department for BioMedical Research (DBMR), University of Bern, 3010 Bern, Switzerland

**Keywords:** hyperuricemia, uric acid, preeclampsia, glucose transporter 9, knockout model

## Abstract

Hyperuricemia is a common feature in pregnancies compromised by pre-eclampsia, a pregnancy disease characterized by hypertension and proteinuria. The role of uric acid in the pathogenesis of pre-eclampsia remains largely unclear. The aim of this study was to investigate the effect of elevated uric acid serum levels during pregnancy on maternal blood pressure and neonatal outcome using two different murine knockout models. Non-pregnant liver-specific GLUT9 knockout (LG9KO) mice showed elevated uric acid serum concentrations but no hypertensive blood pressure levels. During pregnancy, however, blood pressure levels of these animals increased in the second and third trimester, and circadian blood pressure dipping was severely altered when compared to non-pregnant LG9KO mice. The impact of hyperuricemia on fetal development was investigated using a systemic GLUT9 knockout (G9KO) mouse model. Fetal hyperuricemia caused distinctive renal tissue injuries and, subsequently an impaired neonatal growth pattern. These findings provide strong evidence that hyperuricemia plays a major role in the pathogenesis of hypertensive pregnancy disorders such as pre-eclampsia. These novel insights may enable the development of preventive and therapeutic strategies for hyperuricemia-related diseases.

## 1. Introduction

Preeclampsia, characterized by hypertension and proteinuria, is a multisystemic disorder affecting 2–8% of all pregnancies worldwide [1,2]. Contributing substantially to maternal and fetal morbidity and mortality pre-eclampsia represents a heavy psychological and socioeconomic burden for society [3]. Hyperuricemia, a typical feature of patients affected by pre-eclampsia, is one of the earliest and most consistent laboratory findings in pre-eclampsia [4]. Initial studies suggested that hyperuricemia in pre-eclampsia may be based on inflammatory processes and altered renal clearance. There is a growing body of evidence, however, that uric acid plays a role in the pathogenesis of pre-eclampsia [4,5]. Further it has been suggested that hyperuricemia is at least equally important as proteinuria for the assessment of fetal and maternal risk [6,7]. However, its role as a biochemical marker remains disputed [8]. 

In humans and higher apes, uric acid is the end product of nitrogen metabolism due to mutational silencing of the liver enzyme uricase, which is responsible for the oxidation of uric acid to allantoin in other mammalian species. This leads to substantially higher uric acid serum levels in humans compared with the majority of mammals [9]. Uric acid acts a potent antioxidant. Under hypoxic conditions or in higher concentration, however, uric acid causes multiple complications such as gout disease, kidney stone formation, metabolic diseases, cardiovascular complications and hypertension [10,11,12,13,14,15]. Serum uric acid levels decrease in the first weeks of pregnancy, then increase, most likely due to fetal and placental uric acid production and decrease towards the end of the pregnancy [16]. However, in pregnancies complicated by pre-eclampsia, elevated uric acid levels are commonly found [4]. Hyperuricemia is usually considered secondary to altered kidney function, but several studies have shown a link between the severity of the disease and uric acid concentration [17,18,19] as well as uric acid concentration and fetal outcome [19,20].

The kidney is the major player in uric acid homeostasis. About 90% of the uric acid secreted by the renal glomeruli is reabsorbed by the renal tubuli. GLUT9 belongs to the glucose transporter family SLC2, which mediates the transport of small carbon molecules across the membranes in various organs. Compared to the other glucose transporter family members, GLUT9 is different since its primary transport substrate is not glucose/fructose but uric acid [21,22]. GLUT9 was found to be expressed in the kidney, liver, placenta, leukocytes, and chondrocytes [23] and has two primary sub-isoforms, GLUT9a and GLUT9b. The only difference between these two splice-variants is the N-terminal domain [24,25]. We have previously reported that the N-terminal domain of GLUT9a has a regulatory function for iodine [26]. In the kidney, GLUT 9a is expressed on the basolateral side of the proximal tubule, while GLUT9b is expressed on the apical side of the collecting duct [27]. Placental GLUT9a and GLUT9b however co-localize with the villous (apical) membrane but not with the basal membrane of the syncytiotrophoblast [28]. The simultaneous presence of GLUT9a and GLUT9b in the microvillus membrane of syncytiotrophoblasts may enhance clearing the syncytial epithelium and, in turn, facilitate the uric acid transport from the fetoplacental unit into the maternal circulation. 

The aim of the present study was to investigate the effect of elevated uric acid serum levels on blood pressure during pregnancy. For this purpose, we used a liver-specific GLUT9 knockout (LG9KO) mouse model, which in combination with inosine supplementation leads to severe hyperuricemia. Further, we used the systemic GLUT9 knockout (G9KO) mouse model to investigate the postnatal development of fetuses exposed to higher uric acid concentrations.

## 2. Materials and Methods

### 2.1. LG9KO Mice

LG9KO mice were a gift from Professor Bernard Thorens (Center for Integrative Genomics, University of Lausanne, Lausanne, Switzerland) and were characterized earlier [29]. All breedings and colony maintenance, and animal experimentation were performed in the local animal facility under protocols approved by the authorities. Mice were housed with a maximum of 5 animals per cage at 23 °C, with ad libitum access to water and food and a 12 h day/night cycle. At 10 weeks of age, all female LG9KO and control mice were injected intraperitoneally (i.p.) once daily for three consecutive days with tamoxifen (1 mg/mouse) and 100 μL of 10 mg/mL 1:10 EtOH:sunflower oil (Sigma-Aldrich, Buchs, Switzerland) to induce liver-specific GLUT9 gene inactivation. The mice were mated one week after the last injection. The animal experiments were permitted by the Ethical Committee of the Canton of Bern, Switzerland (Kantonale Ethikkommission Bern; #BE100/14).

### 2.2. Telemetric Measurements

Pressure transmitters (DSI PhysioTel PA-C10, DSI, Saint Paul, MN, USA) were implanted in mice under anesthesia (isoflurane). The catheter tip was introduced into the carotid artery and positioned in the aortic arch and the implant secured subcutaneously on the flank. Mice were allowed to recover for 2 weeks. The DSI acquisition system (DSI) with 16 receiver plates was used for simultaneous measurements of heart rates and mean arterial blood pressures. Two cohorts of mice were measured in the study. Heart rate and mean arterial blood pressure were measured throughout the entire gestation. During gestation, all mice received inosine-supplemented diets. Only mice showing a proper constant signal transmission during the entire pregnancy were analyzed. 

### 2.3. G9KO Mice

G9KO mice were also obtained from Professor Bernard Thorens (Center for Integrative Genomics, University of Lausanne, Lausanne, Switzerland) and described earlier [30]. Heterozygous animals were crossbred, which resulted in wild-type (WT) and knockout (KO) pups in the same litter. Pregnant mice were fed with either standard chow alone or (to increase uric acid serum levels) with supplementation of 1 g/kg inosine after mating. Following delivery, the diet of all pups was standard chow. The pups’ body weights were assessed daily up to postnatal day 70. At day 70, animals were anesthetized and perfused with 20 mL PBS through the right heart ventricle, followed by perfusion with 4% paraformaldehyde (PFA) in PBS. Organs were dissected, weighed, and fixed in 4% PFA for histological analysis. The animal experiments were permitted by the Ethical Committee of the Canton of Bern, Switzerland (Kantonale Ethikkommission Bern; #BE100/14).

### 2.4. Immunohistochemistry

Mouse kidneys were fixed in formaldehyde solution (4% (*v*/*v*); Merck, Whitehouse Station, NJ, USA) for 2–4 h at room temperature (RT) followed by 4 °C for a total time of 24–48 h. Fixed kidney and placentae were embedded in paraffin, sectioned into 3 μm coronal slices, deparaffinized (xylene, 2 × 5 min) and rehydrated (100% ethanol, 2 × 3 min; 95% ethanol, 1 min; 70% ethanol, 1 min; rinsed in distilled water). After deparaffinization and rehydration of the slices, the target was retrieved in Tris-EDTA buffer (10 mM Tris-Base, 1 mM EDTA, 0.05% (*w*/*v*) Tween 20, pH 9.0) by heat treatment in a pressure cooker for 15 min. Slides were washed in PBS and Tween 20 0.1% (*w*/*v*) and blocked with goat serum 10% (*v*/*v*), BSA 1% (*w*/*v*) in PBS. Then, the slides were stained with antibodies against human F4/80 (1:100, abcam ab6640) and CD3 (1:100, abcam ab16669) before they were counterstained with either DAB or Mayer’s Hematoxilin Solution, the Dako Cytomation EnVision System-HRP (Dako, Glostrup, Denmark). The slides were washed in PBS and Tween20 0.1% (*w*/*v*, 2 × 5 min) and incubated with the endogenous peroxidase block solution (Dako S2003) for 15 min at RT. Peroxidase-labeled polymer was applied to the slides for 30 min at RT, followed by 3 washes in PBS (5 min each) and the addition of 3,3′-diaminobenzidine in chromogen solution in buffer substrate (SigmaD 3939) for 10–30 min, according to the manufacturer’s instructions. Slides were rinsed in H2O and counterstained with hematoxylin and eosin (HE) (Fluka, Switzerland) for 2 min and then rinsed with tap water for 1 min, dehydrated in a series of ethanol baths (70%, 95%, 100%, *v*/*v*) and xylene and mounted with Eukitt (Sigma-Aldrich, St. Louis, MO, USA). 

### 2.5. Statistical Analysis

The investigators performing the statistical analysis were blinded. Data are expressed as mean ± SEM. Graph prism software was used for statistical analysis. Differences between means were tested using Student’s *t*-test or repeated measures ANOVA where appropriate. A *p*-value of <0.05 was considered to reach significance.

## 3. Results

### 3.1. Gestational Hyperuricemia Leads to Elevated Blood Pressure and Renal Injury

To address whether hyperuricemia leads to a preeclampsia-like phenotype, we used the LG9KO mouse model, which lacks GLUT9 expression in the liver and therefore prevents degradation by uricase. Challenging the pregnant LG9KO mice with inosine food supplementation increased uric acid levels up to 195 ± 14 µmol/L compared to wild-type animals (82 ± 8 µmol/L, Appendix A). In the first gestational week, corresponding to the first trimester of human pregnancy, wild-type (WT) mice and LG9KO mice showed similar mean arterial pressure (MAP) values (Figure 1A). During the second gestational week, blood pressure dropped in both groups, which occurs commonly during the second trimester in humans as well as in mice [28]. This drop, however, was smaller in the LG9KO mice than in WT mice, leading to significantly higher blood pressure values in the LG9KO mice compared with WT mice. The discrepancy persisted during the second and third trimesters of pregnancy when blood pressure rose in both groups. Figure 1B depicts the peripartal blood pressure values, which were substantially higher in the LG9KO mice than in their WT peers. At birth, the blood pressures of LG9KO and WT mice were 126 ± 4 and 106 ± 4 mmHg, respectively (Student’s *t*-test, *p* < 0.001, Figure 1B). Immediately following birth, WT mice showed blood pressure values similar to preconception and first-trimester measurements, while LG9KO mice showed higher blood pressure values than WT mice until the fourth postpartum day (Figure 1A).

To analyze the circadian blood pressure dipping, 3-hour averages in the middle of the night (active phase) and day (resting phase) were compared in both groups. Both WT and LG9KO showed dipping at the beginning of pregnancy (Figure 2). Nine days before birth, LG9KO animals lost their dipping, which was restored at the fifth postpartum day, while WT mice lost their dipping only 3 days before birth until postnatal day 3. A panel of all single days is shown in Appendix A.

To assess the potential impact of hyperuricemia on maternal outcome, histologic examination of the kidneys was performed following sacrifice of the animals on postnatal day 6. F4/80-staining of renal tissue obtained from LG9KO animals showed distinct signals reflecting macrophage invasion as a sign of inflammation, while no immunohistochemistry signals were detected in WT mice (Appendix A). 

### 3.2. Fetal Hyperuricemia Leads to Postnatal Growth Restriction

To evaluate the impact of fetal hyperuricemia on fetal and/or postnatal development, we used a systemic G9KO animal model lacking GLUT9 expression in all fetal organs, including the liver and the placenta. Heterozygous mice were crossed, and the pups were genotyped. In the 422 offspring, no change in the Mendelian ratio could be observed (WT: *n* = 103, heterozygous: *n* = 221, KO: *n* = 98 animals), indicating no increased fetal lethality due to a systemic G9KO. Uric acid levels in G9KO homozygous fetuses were 5.3-fold and 4.7-fold higher than in WT or heterozygous animals, respectively (homozygous G9KO 142 ± 6, WT 27 ± 2, heterozygous G9KO 30 ± 2 mmol/L mean ± SEM, Figure 3A). Further homozygous G9KO fetuses showed a 5-fold higher uric acid serum concentration than their mothers (142 ± 6 versus 28 ± 3 mmol/L, *p* < 0.0001). Heterozygous and WT fetuses did not differ in their uric acid levels compared to their mothers (heterozygous or WT fetuses versus mother animals: 30 ± 2 or 27 ± 2 versus 28.3 mmol/L, respectively, *p* > 0.05 and *p* > 0.05).

To evaluate the offspring’s growth pattern, body weight was measured from postnatal day 7 until day 70, and a histologic examination was performed following sacrifice. While the female KO offspring showed a significantly lower body weight during the accelerated growth period (postnatal day 19 until day 40) compared to the WT peers (Figure 3B), daily weight measurements revealed no difference between WT, heterozygous, and KO male animals (Figure 3C).

To assess whether growth patterns are modulated by different degrees of hyperuricemia, pregnancies were challenged by inosine food supplementation to increase uric acid levels in the fetal circulation. Following chow with inosine, uric acid levels in WT, heterozygous, and KO fetuses were increased up to 46 ± 3, 61 ± 3, and 195 ± 8 (mean ± SEM) mmol/L, respectively (Figure 3A). There was a 4.2-fold increase in uric acid serum levels in homozygous G9KO fetuses when compared with WT peers (homozygous G9KO versus WT: 195 ± 8 versus 46 ± 3, *p* < 0.0001), whereas no differences in serum levels were observed between heterozygous G9KO or WT fetuses and those of their mothers (WT or heterozygous G9KO fetuses versus mother animals: 61 ± 3 or 46 ± 3 versus 48 ± 4, *p* > 0.05 and *p* > 0.05). Homozygous G9KO fetuses showed a 4-fold higher uric acid serum concentration than their mothers (homozygous G9KO fetuses versus mother animals: 195 ± 8 versus 48 ± 4, *p* < 0.0001).

Interestingly, upon inosine supplementation, body weight gain in male offspring was reduced in homozygous G9KO pups compared to WT controls (Figure 3C), while in female pups, the reduction in body weight gain observed in homozygous G9KO mice at days 19–40 was not further aggravated by inosine supplementation (Figure 3B). Regarding the body weight gain, the female homozygous G9KO offspring were able to catch up with their WT mates, whereas the male homozygous G9KO offspring showed substantial neonatal growth retardation from day 18–70, yielding a difference of 13% in body weight at postnatal day 70.

### 3.3. Fetal Hyperuricemia Causes Renal Injury

To assess tissue damage upon hyperuricemia, the kidneys of the offspring were macroscopically and histologically examined. In females, kidneys obtained from homozygous G9KO show a wizened surface (Figure 4A) and less weight than those from WT following a diet with or without inosine (Figure 4B). Compared to controls, female homozygous KO offspring showed a kidney weight reduction of 15% under the standard chow diet, whereas homozygous KO mice showed a reduction of 28% under the inosine diet. Kidneys of homozygous G9KO offspring revealed distinct hydronephrosis as well as renal cellular swelling, tubular atrophy, and interstitial fibrosis predominantly in the superficial cortex (Figure 4C). Compared to WT mice, homozygous G9KO peers showed a distinct immunohistological signal for F4/80 and CD3, revealing macrophage and T-cell invasions reflecting signs of chronic inflammation in fibrotic areas (Figure 4C). Following the challenge with the inosine diet, F4/80 macrophage signals in the homozygous G9KO offspring were 3.7-fold higher than in controls (homozygous G9KO versus WT: 10.8 ± 0.9 versus 2.9 ± 0.2, arbitrary numbers, *p* < 0.001, Appendix A).

Similarly, homozygous G9KO male kidneys weighed less than those from their WT peers following a diet with or without inosine (Figure 5A): Compared to control males, homozygous KO offspring showed a kidney weight reduction of 18% under the standard chow diet and reduction of 46% under inosine diet (Figure 5B). These kidneys were also characterized by hydronephrosis and by altered histological findings such as tubular atrophy and interstitial fibrosis in the superficial cortex. Immunohistological staining for F4/80 and CD3 showed a marked chronic inflammation, which was even more pronounced than in females (Figure 5C). Following the challenge with the inosine diet, the macrophage signals in the homozygous G9KO offspring were 4.0-fold higher than in controls (homozygous G9KO versus WT: 13.6 ± 2.6 versus 3.4 ± 0.4, arbitrary numbers, *p* < 0.01, Appendix A).

To determine whether these observed damages appear already prenatally, fetuses were sacrificed at day 18.5 following cesarean section, and their kidneys were analyzed histologically. Already prenatally, G9KO fetuses exhibit distinct inflammation signs, such as disturbed nephron formation and substantial swelling of renal cells, resulting in a reduction in the nephron lumen (Appendix A).

## 4. Discussion

Our data demonstrate that in murine pregnancies compromised by maternal hyperuricemia, blood pressure levels increase during the second and third trimesters compared to controls. Moreover, these hypertensive mice lose their circadian blood pressure dipping during the third trimester, while WT mice omit the dipping only for three prenatal and three postnatal days. In our animal model, fetuses exposed to hyperuricemia during pregnancy were already prenatally affected by distinct renal tissue injuries. Moreover, offspring following pregnancies upon fetal hyperuricemia were compromised by neonatal growth retardation and by substantial renal lesions.

These findings demonstrate that elevated maternal uric acid serum levels during pregnancies lead to increased blood pressure and loss of circadian blood pressure dipping, at least in our rodent model. These clinical signs are typical features observed in human pregnancies affected by pre-eclampsia. In normal pregnancies, blood pressure levels decrease in the first trimester, whereas in pre-eclampsia, blood pressure levels remain stable during the first trimester and increase continuously during the second and third trimesters [31]. Of note, several studies reported decreased circadian blood pressure dipping in human pregnancies affected by hypertensive pregnancy disorders such as pre-eclampsia [31,32]. In our pregnant mice with hyperuricemia, the rise of blood pressure and loss of blood pressure dipping started at 9 days of gestational age. Since a murine pregnancy lasts 21 days, the occurrence of these clinical signs is comparable with pre-eclampsia, which evolves in the second half of human pregnancy. The highest differences with up to 20mmHg in MAP between maternal animals with and without hyperuricemia were observed around birth. This dynamic pattern of altered blood pressure levels in pregnant mice under hyperuricemia parallels human gestation.

Our findings are of great interest since, upon hyperuricemia, LG9KO mice do not exhibit hypertension while not pregnant. Even by challenging these animals with a considerable amount of inosine supplementation (15 g/kg per chow diet), reaching uric acid levels up to 300 µmol/l, they do not alter their blood pressure levels, even though they exhibit severe renal tissue damage [29]. In our study setting, however, pregnant mice fed with 15-fold less inosine (i.e., 1 g/kg) yielding uric acid concentrations of 195 µmol/l show a distinct phenotype with elevated blood pressure levels and loss of dipping, which are typical features of (human) pregnancies affected by pre-eclampsia. These findings corroborate our hypothesis that hyperuricemia or pregnancy per se do not lead to elevated blood pressure levels, whereas the coincidence triggers the development of hypertension. Although an association between uric acid and an increased risk for adverse pregnancy outcomes, including hypertensive pregnancy disorders, has been known for more than a century [19,33,34], the exact causative relationship remained largely elusive. Based on our data, we have solid evidence that uric acid plays an important role in the pathogenesis of hypertensive pregnancy disorders in our animal model. 

Furthermore, following pregnancies complicated by hyperuricemia, the kidneys of our maternal hypertensive mice showed distinct end-organ damage. Similarly, acute kidney disease is the most common renal complication in human pregnancies compromised by pre-eclampsia [35]. Moreover, loss of circadian blood pressure dipping is associated with severe end-organ damage and aggravation of cardiovascular risk [36]. The clinical relevance of hypertension during pregnancy lies in an increased risk for metabolic changes and cardiovascular diseases later in life [37]. The common clinical and pathohistological features highlight that LG9KO mice may represent a useful model to investigate the underlying pathomechanisms leading to hypertensive pregnancy disorders.

To elucidate the impact of hyperuricemia on the perinatal outcome, we used a G9KO mice model in which both the fetus and its placenta lack GLUT9. Since uric acid has no access to the liver enzyme uricase and, in turn, cannot be metabolized in G9KO fetuses, the only way to eliminate uric acid from fetal circulation is via transplacental transport. We have previously shown that the placenta efficiently maintains uric acid equilibrium between maternal and fetal circulation [28]. Placental GLUT9 plays a major role in fetal uric acid homeostasis since G9KO fetuses (lacking hepatic and placental GLUT9) show substantially higher uric acid serum levels than their mothers. We hypothesized that fetal exposure to elevated uric acid levels might also have deleterious effects on renal development. Therefore, we aimed to assess the impact of fetal hyperuricemia on kidney development. Kidney examination of 70-day-old offspring prenatally exposed to hyperuricemia revealed macro- and microscopical changes representing severe chronic kidney injury. These end-organ damages were (at least partly) already prenatally detectable in hyperuricemia-exposed fetuses (Appendix A), indicating that hyperuricemia impairs renal organogenesis. 

Any increased inflammatory state during early pregnancy may facilitate the development of pre-eclampsia [38]. This concept is corroborated by preventive strategies with anti-inflammatory agents, such as aspirin, showing a beneficial impact on pregnancy outcomes [39]. Resveratrol, another agent with anti-inflammatory properties, was shown to ameliorate an oxidative-stress reaction in a rodent pregnancy model with hypertension and proteinuria [40] and might stimulate the invasive capability of human trophoblasts by promoting EMT and mediating the Wnt/β-catenin pathway in PE [41]. It is plausible that uric acid is involved in these processes since resveratrol was shown to lower uric acid serum levels in mice compromised by hyperuricemia [42]. The question of whether resveratrol directly regulates the GLUT9-mediated uric acid transport or acts via other pathways remains to be elucidated. 

To address the question of whether the observed renal changes impact postnatal development, we assessed the growth patterns of prenatally hyperuricemia-exposed offspring. Upon fetal hyperuricemia with a mean of 142 ± 6 µmol/L uric acid, postnatal growth was compromised in females, whereas growth patterns in males stayed unaffected. Increased mean uric acid levels of 195 ± 12 µmol/L, however, lead to postnatal growth retardation in males, while females were not affected by growth retardation when compared with their peers exposed to moderate uric acid levels of 142 ± 6 µmol/L. These findings indicate that female offspring seem more vulnerable to mild hyperuricemia than male offspring. Following initial growth restriction, however, females tend to accelerate their body weight gain. We speculate that this accelerated growth pattern may be due to the sequelae of prenatal exposure to elevated uric acid levels. Indeed, children born following pregnancies affected by pre-eclampsia show metabolic and cardiovascular alterations such as increased BMI and elevated blood pressure [43], a concept known as fetal programming [44,45]. 

In summary, our data provide strong evidence that hyperuricemia plays an important role in the pathogenesis of hypertensive pregnancy disorders, at least in our rodent model. Moreover, our findings indicate that hyperuricemia during pregnancy has a deleterious impact on fetal renal development and adequate neonatal growth pattern. Similarly, also in human pregnancies, hyperuricemia may trigger the development of pre-eclampsia and impacts fetal programming. Hence, we encourage further animal and clinical studies to investigate the role of uric acid in the pathogenesis of hypertensive pregnancy disorders. The gaind insights may help develop novel therapeutic and preventive strategies for hyperuricemia-related (pregnancy) disorders such as pre-eclampsia, gout, and metabolic syndrome.

## Figures and Tables

**Figure 1 cells-11-03703-f001:**
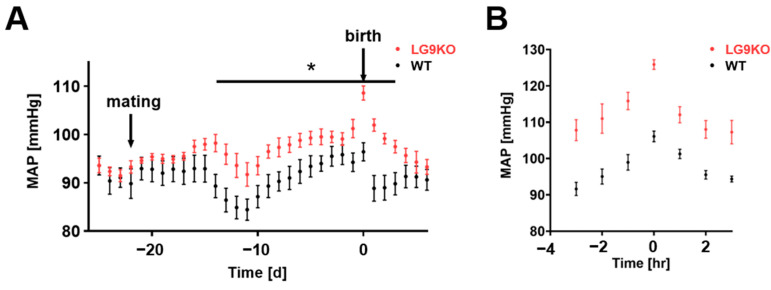
Time course of blood pressure values with and without hyperuricemia: (**A**) Blood pressure diagram of WT (black, *n* = 7) and LG9KO (red, *n* = 8) animals during gestation. Each point represents a 24 h average of the mean arterial pressure ± SEM. At day eight of gestation, the mean arterial pressure of the LG9KO animals’ increases compared to the WT group; this difference disappears at day 5 after birth. (**B**) One-hour averages (indicated timepoint ± 30 min) of peripartal blood pressure. A difference of 19 mmHg was observed between LG9KO and WT groups at the time of birth (t = 0). * *p* < 0.05.

**Figure 2 cells-11-03703-f002:**
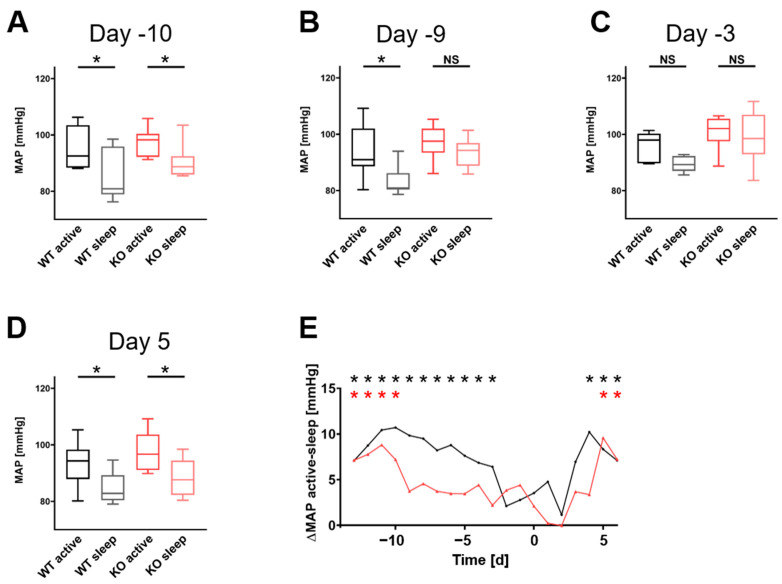
Blood pressure dipping: (**A**–**D**) are dipping plots at days −10, −9, −3, and +5. The LG9KO animals (red) lose the dipping at day −9, and it returns at day 5, while the WT animals (black) lose the dipping at day −3, returning on day 3 (Appendix A). (**E**) ΔMAP of MAPactive–MAPsleep from day −13 to day 6. LG9KO animals (red) lose their dipping much earlier than the WT animals (black). * statistically significant dipping (passive vs. active), *p* < 0.05. NS, not significant.

**Figure 3 cells-11-03703-f003:**
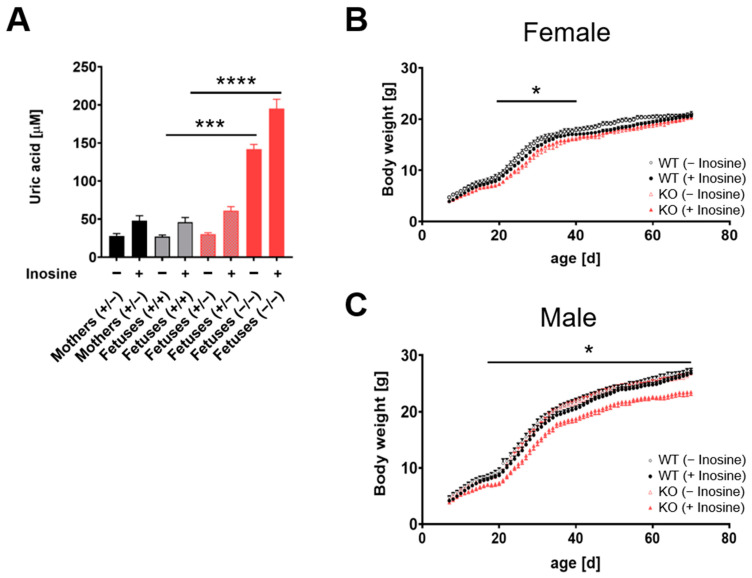
Fetal hyperuricemia and growth pattern: (**A**) Serum uric acid levels at day 18.5 of the pregnant mice fed with standard chow and fetuses of all three genotypes. To increase the KO fetuses’ uric acid levels, the pregnant dams’ chow was supplemented with 1 mg/g inosine during the whole gestation. Inosine supplementation further increases the uric acid level of KO animals. (**B**,**C**) Analysis of weight development of female (**B**) and male (**C**) pups, under control conditions (circles) and inosine supplementation (triangles) during gestation. WT (female *n* = 7, male *n* = 7) are shown in black, and KO animals (female *n* = 5, male *n* = 8) in red. Data represent mean ± SEM. In female animals (**B**), a statistical significance could be observed during days 19–40 after birth between KO and WT animals without inosine supplementation. No statistical difference could be observed between KO animals with and without inosine. In male animals (**C**), only KO animals exposed to high uric acid concentration during fetal development show a significant difference in body weight from days 18–70 after birth compared to WT controls (+inosine) (female *n* = 10, male *n* = 10). No difference could be observed in KO animals (female *n* = 10, male *n* = 7) exposed to lower uric acid concentration during fetal development when compared to WT controls (−inosine). * *p* < 0.05, *** *p* < 0.001, **** *p* < 0.0001.

**Figure 4 cells-11-03703-f004:**
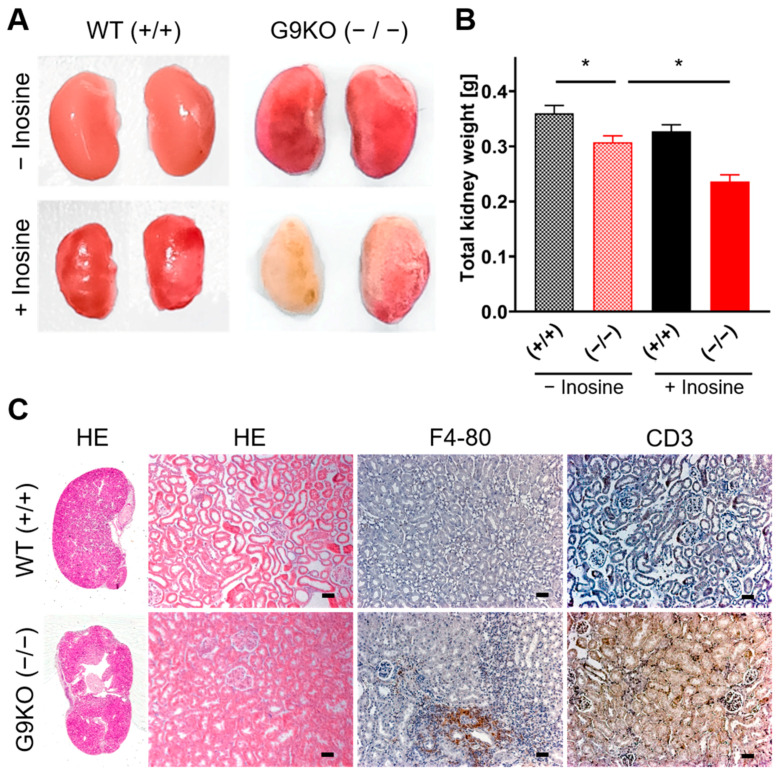
Female kidney status: (**A**) Kidneys isolated at day 70 after birth from WT and G9KO female mice under a control diet and inosine supplementation. Mice show severe morphological changes, which worsen under inosine supplementation. (**B**) Summary of total kidney weight (mean ± SEM) in WT (+/+) and KO (−/−) females. Inosine supplementation further reduces kidney weight in G9KO animals. (**C**) HE, F4/80, and CD3 stainings of WT (+/+) and KO (−/−) animals exposed to inosine during fetal development. Scale bar: 25 µm; * *p* < 0.05.

**Figure 5 cells-11-03703-f005:**
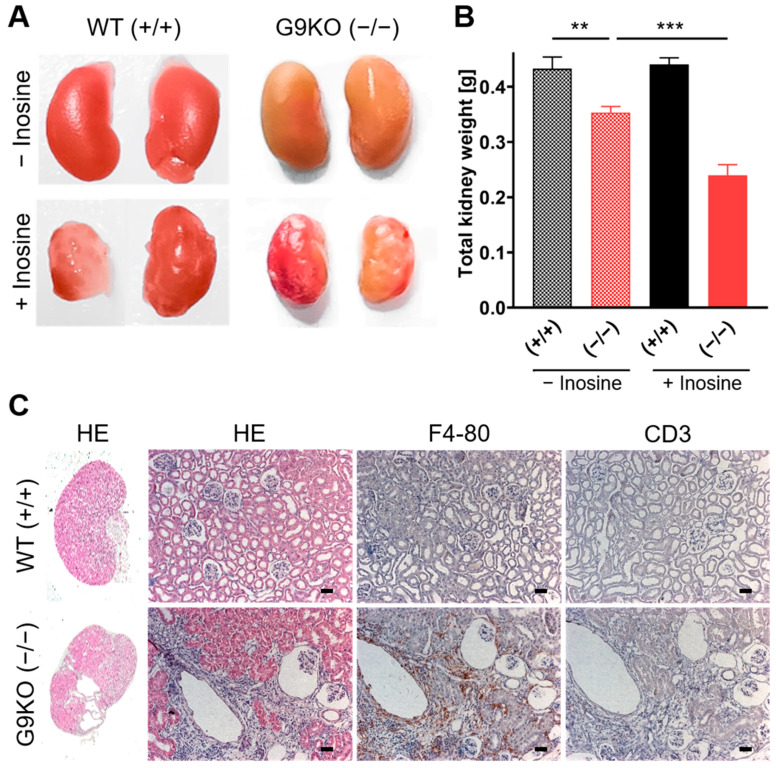
Male kidney status: (**A**) Kidneys isolated at day 70 after birth from WT and G9KO male mice under a control diet and inosine supplementation. The morphological changes observed in G9KO male animals are significantly higher than the morphological changes observed in female G9KO animals. (**B**) Inosine supplementation results in a higher reduction in kidney mass in KO compared to WT animals. (**C**) HE, F4/80 (macrophages), and CD3 (T cells) stainings of WT (+/+) and KO (−/−) animals exposed to inosine during fetal development. Scale bar: 25 µm size; ** *p* < 0.01, *** *p* < 0.001.

## Data Availability

All data are provided by the authors on request.

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
