# Peer review of "Hyperuricemia during Pregnancy Leads to a Preeclampsia-Like Phenotype in Mice"

_cells, 2022, doi:10.3390/cells11223703_

Round 1
Reviewer 1 Report
the manuscript is interesting but image quality is very low and must be improved before reviewing. Authors should consider the idea to put IHC images in dedicate figures.
Figures must be shown in the paragraph when they are mentioned and not at the end of results.
Figures 1B and 2C are not visible
Figure 4C and 5C: an higher magnification is necessary to see the tissue morphology and specific staining
Figure S2 and S4 are unreadable
Author Response
The figures have been optimized and enlarged to cover the whole page width and are now inserted after the paragraph in which they are first mentioned. Some relatively small characters have been increased in size for better readability. The Supplementary Figures are now included as a separate file. A graphical abstract has been added for a better overview.
Reviewer 2 Report
The studies on animals are fundal in perinatal medicine since sixtees, as many examinations are not application able to human fetus.
The aim of the study was to investigate the effect of elevated uric acid serum levels on blood pressure during pregnancy. For this purpose, mouse model has been used.
Material and methods are well and clearly described.
Statistical methods are well applied.
Results that come out from the research are the consequence of the studied subject:
1. Gestational hyperuricemia leads to elevated blood pressure and renal injury
2. Fetal hyperuricemia leads to postnatal growth restriction
3. Fetal hyperuricemia causes renal injury
Results are supported by several graphs and photos, which, in my opinion should be more readible and bigger.
Disscussion is led as a consequence of results of the present study and consists of new hypotheses.
In my opinion the paper is worth to be published in Cells.
Author Response
The figures have been optimized and are now inserted after the paragraph in which they are first mentioned. Some relatively small characters have been increased in size for better readability. The Supplementary Figures are now included as a separate file. A graphical abstract has been added for a better overview.
Round 2
Reviewer 1 Report
the manuscript has been significantly improved and can be accepted in the present form